# A CMOS Optoelectronic Transceiver with Concurrent Automatic Power Control for Short-Range LiDAR Sensors

**DOI:** 10.3390/s25030753

**Published:** 2025-01-26

**Authors:** Yejin Choi, Juntong Li, Dukyoo Jung, Seonhan Choi, Sung-Min Park

**Affiliations:** 1Division of Electronic & Semiconductor Engineering, Ewha Womans University, Seoul 03760, Republic of Korea; enun0515@ewha.ac.kr (Y.C.); lijuntong@ewhain.net (J.L.); seonhan.choi@ewha.ac.kr (S.C.); 2Graduate Program in Smart Factory, Ewha Womans University, Seoul 03760, Republic of Korea; 3College of Nursing, Ewha Womans University, Seoul 03760, Republic of Korea; dyjung@ewha.ac.kr

**Keywords:** APC, APD, LiDAR, optoelectronic, sensor, transceiver, VCSEL

## Abstract

This paper presents an optoelectronic transceiver (OTRx) realized in a 180 nm CMOS technology for applications of short-range LiDAR sensors, in which a modified current-mode single-ended VCSEL driver (m-CMVD) is exploited as a transmitter (Tx) and a voltage-mode fully differential transimpedance amplifier (FD-TIA) is employed as a receiver (Rx). Especially for Tx, a concurrent automatic power control (APC) circuit is incorporated to compensate for the inevitable increase in the threshold current in a VCSEL diode. For Rx, two on-chip spatially modulated P^+^/N- well avalanche photodiodes (APDs) are integrated with the FD-TIA to achieve circuit symmetry. Also, an extra APD is added to facilitate the APC operations in Tx, i.e., concurrently adjusting the bias current of the VCSEL diode by the action of the newly proposed APC path in Rx. Measured results of test chips demonstrate that the proposed OTRx causes the DC bias current to increase from 0.93 mA to 1.42 mA as the input current decreases from 250 µA_pp_ to 3 µA_pp_, highlighting its suitability for short-range sensor applications utilizing a cost-effective CMOS process.

## 1. Introduction

In recent decades, light detection and ranging (LiDAR) technology has been gaining a great deal of attention across various application fields [1,2,3,4,5]. A typical LiDAR sensor utilizes a laser diode (LD) in a transmitter (Tx) to emit light pulse signals, and an avalanche photodiode (APD) in a receiver (Rx) to analyze the reflected signals from a target, thereby accurately extracting crucial information such as distance, speed, and depth. Therefore, LiDAR sensors can play a pivotal role in applications such as autonomous driving, environmental monitoring, 3D mapping, and home monitoring systems.

Figure 1 depicts the block diagram of a typical LiDAR sensor [6,7], where the Tx primarily consists of a laser diode and its associated driving circuits for emitting light signals toward target objects. The Tx incorporates an automatic power control (APC) module to stabilize the output power of the employed LD, regardless of temperature variations or aging effects of the LD. Specifically, the APC monitors the laser’s output power through a feedback loop (with a monitoring photodiode or MPD) and adjusts the bias current accordingly.

Previous research has primarily focused on LD drivers for long-range LiDAR systems that require narrow, high-power optical pulses [8]. As an example, Ref. [9] introduced a high repetition rate CMOS driver for generating high-energy sub-nanosecond laser pulses in SPAD-based time-of-flight range finding. Although these advancements have significantly improved the LD driver design, laser diodes often face challenges due to high costs and substantial bias voltage requirements. Vertical-cavity-surface-emitting laser (VCSEL) diodes, in contrast, offer a more affordable and energy-efficient alternative with benefits such as low bias voltage and cost. As highlighted in Ref. [10], VCSELs have been adopted in cost-sensitive applications.

Yet, it is well known that VCSEL diodes are unidirectional and thus mandate the use of an optical splitter (or coupler) to monitor the emitted optical power of APC operations. Hence, a novel optoelectronic transceiver (OTRx) is proposed in this work, which exploits an on-chip APD in Rx to control the bias current of the utilized VCSEL diode in Tx. Figure 2 shows the architecture of the proposed OTRx, in which the Tx employs a modified current-mode single-ended VCSEL driver (m-CMVD) as a front-end circuit, whereas the Rx exploits a voltage-mode, fully differential transimpedance amplifier (FD-TIA) with two on-chip APDs. The m-CMVD in Tx ensures reliable and consistent optical performance and effectively compensates for the well-known variations caused by temperature and device aging [11]. Meanwhile, the analog front-end (AFE) circuit in Rx, i.e., the FD-TIA, amplifies weak reflected signals and filters out noise, thus maximizing signal-to-noise ratio (SNR) to enable precise distance and depth measurements.

## 2. Circuit Description

Figure 3 depicts a block diagram of the m-CMVD in Tx that integrates a wide-swing current–mirror circuit as a bias generator, thereby providing stable bias current. The m-CMVD ensures reliable and consistent optical performance. In this work, the wide-swing current mirror produces a 1 mA reference current.

However, the threshold current of VCSEL diodes may increase due to temperature variation. This increase in threshold current can lead to performance degradation issues, such as amplitude variation, increased timing jitter, and reduced bandwidth. These issues can ultimately deteriorate the reliability of VCSEL diodes. An automatic power control (APC) mechanism is employed to mitigate this issue in general, thereby compensating for the temperature-induced rise in threshold current. Therefore, the APC circuit monitors the output power of a VCSEL diode by utilizing a feedback loop and adjusts the bias current accordingly to stabilize the output power.

The proposed APC circuit in this work, as shown in Figure 2, comprises an on-chip APD with its cathode grounded, a narrow-bandwidth shunt-feedback TIA with a low-pass filter to extract the DC voltages, and a comparator with a reference voltage to activate the APC operations. Therefore, the reflected optical pulses from target objects enter this APC circuit to provide information regarding any reduction in the bias current of the VCSEL diode directly, and to compensate the bias current dynamically for the stable operations of Tx.

Meanwhile, the modulation current of m-CMVD (max. 10 mA_pp_) is derived in the modified current-steering logic (m-CSL) with the input voltage (V2) via an input buffer (IB). The m-CSL is concurrently biased by the DC current generated from the bias generator and is either turned on or off by the amplitudes of the input voltage (V2), thereby passing the desired modulation current into the external VCSEL diode [12].

As shown in Figure 4a, I_BIAS_ is sourced from the bias generator, where the gate-source voltage (V_gs6_) of M6 is set to closely match the threshold voltage. This ensures a stable generation of the reference current (I_REF_). Then, this I_REF_ is mirrored through the NMOS transistors (M8, M9) in the m-CSL circuit (shown in Figure 4b). Both the modulation current (I_MOD_) and the bias current (I_BIAS_) are generated and controlled through the current–mirror configuration. Here, it should be noted that the aspect ratio (W/L) of the NMOS transistors (M8, M9) is carefully adjusted to provide precise current mirroring. By increasing the W/L ratio, the current-handling capability of the current–mirror transistors is proportionally scaled. For instance, if the W/L ratio of M9 is set to be twice that of M8, the mirrored current through M9 will be approximately twice the reference current flowing through M8. This method provides an efficient and scalable way to adjust the modulation current (I_MOD_) to meet the specific requirements of the VCSEL driver.

Figure 4c depicts the schematic diagram of the IB that uses AND gates and inverters to control the switches (SA2, SA2_b).

This design can effectively replicate the modulation behavior of a VCSEL diode under varying current conditions, while the current mirror’s scalability through the adjusting W/L ratio provides enhanced design flexibility.

Figure 5 illustrates the cross-sectional view of the on-chip P^+^/N-well (P^+^/NW) APD that shares the same structure described in [13]. When off-chip APDs are utilized, signal distortions may occur because of the inevitable bond–wire inductance followed by the parasitic capacitance from the ESD protection diode. It may also cause difficulties in PCB design for testing and increase the implementation cost especially for cases of multi-channel receiver arrays. Therefore, we have eliminated these effects in this work by exploiting on-chip APDs, simultaneously offering merits such as small size, low cost, and facile design.

When light enters the optical window of the APD, electron–hole pairs are generated at the junction between the P^+^ region and the N-well. The generated holes, under the influence of the reverse bias voltage applied to the N-well, create additional electron–hole pairs, thereby triggering the avalanche effect [14]. Then, the generated photocurrents flow into the input node of the FD-TIA that is connected to the central P^+^ contact of the optical window. The design features an octagonal shape to prevent premature edge breakdown [15]. The optical window has a size of 40 × 40 µm^2^. Fabricated in a standard 180 nm CMOS process, the on-chip P^+^/NW APD demonstrates a responsivity of 2.72 A/W at a reverse bias voltage of 11.05 V with a parasitic capacitance of 0.49 pF. The P^+^ contact of the P^+^/NW APD is connected to the input node of the FD-TIA, through which the input photocurrent (i_pd_) flows into the FD-TIA.

The Rx detects the reflected laser pulses and converts them into voltage signals, which are subsequently processed in the following digital circuitry. For this purpose, various topologies have been suggested previously [16,17,18,19,20,21]. Pseudo differential architecture in particular has been frequently exploited for high-speed optical Rx ICs, which insert a passive low-pass filter (LPF) immediately after the front-end TIA to acquire differential signaling [16]. However, this pseudo differential structure requires a multi-stage post-amplifier to generate fully differential outputs and therefore leads to rather large power consumption and a bulky chip area because of the unavoidable passive resistors and capacitors.

As such, in this paper, we propose a fully differential architecture utilizing two on-chip APDs. Figure 6 illustrates the block diagram of the proposed optoelectronic receiver (oRx), where the on-chip APD detects the reflected light from a target and generates an input photocurrent (i_pd_). Also, a dummy APD is added at the differential input node to eliminate the need for a passive capacitor, which would typically be used to electrically model the on-chip APD. This dummy APD helps to mitigate the issue of DC offset currents between the two differential outputs by utilizing a balanced architecture, thus producing symmetrical outputs. In other words, the proposed oRx architecture can eliminate the need for an offset cancelation circuit, consequently reducing the circuit complexity and chip area. The FD-TIA topology converts the input photocurrent to voltage signals and enables the generation of symmetric differential signals even without a multi-stage post-amplifier, consequently reducing power consumption further. In addition, a f_T_-doubler that boosts the output voltages further and matches the output resistance to 50 Ω for facilitating the measurements. Furthermore, this oRx incorporates an extra third APD that is followed by a narrow-bandwidth low-noise TIA. This specially dedicated channel is proposed for the concurrent APC operations.

Figure 7 presents the schematic diagram of the FD-TIA. In particular, the FD-TIA features PMOS loads (M30, M31) that are cross-connected to the source-followers (M32, R5 and M33, R6). Since the gate of each source-follower (SF) is connected to the drain of the PMOS load while the source of each SF is connected to the gate of the PMOS load, the peak voltage of the drain node (OUT−) can become VDD − V_sg30_ + V_gs32_, thus allowing a full swing up to the supply voltage (VDD) under the assumption that the values of V_sg30_ and V_gs32_ are almost identical. According to small signal analysis, the transimpedance gain of the FD-TIA is given by(1)$$\frac{v_{o}^{+}}{i_{pd}}≅\frac{{R_{F}g}_{m28}}{g_{m28}+g_{m30}}≅R_{F}\;\;if\;g_{m28}\gg g_{m30},\;\frac{v_{o}^{-}}{i_{pd}}≅-\frac{{R_{F}g}_{m28}}{g_{m28}+g_{m30}}≅-R_{F}\;if\;g_{m28}\gg g_{m30}$$
where g_mi(i=28,30)_ represents the transconductance of the transistor M_i(i=28,30)_, respectively.

As mentioned previously, there is an extra channel in the oRx specially dedicated to the concurrent APC operations, comprising an on-chip APD with its cathode grounded, a narrow-bandwidth TIA with a LPF, and a comparator. This APC channel receives the optical signals through the extra APD, converts the photocurrent (i_pd_) into a low-frequency (LF) voltage, and then generates a DC bias current if the LF voltage is greater than the reference voltage (VREF). Since the TIA in this APC channel employs a single-ended version of the FD-TIA, its output is converted into a DC level via an LPF consisting of an on-chip R_L_ and an off-chip C_L_. Then, this converted DC level passes through a two-stage operational amplifier and is compared with V_REF_ to produce the desired APC output.

## 3. Layout and Simulation Results

Figure 8 shows the layout of the proposed m-CMVD circuit first, where the fabricated chip occupies a core area of 0.196 mm^2^. Post-layout simulations using a standard 180 nm CMOS process indicate the maximum power consumption of 11 mW from a single 3.3 V supply.

Figure 9 depicts the simulation results of the PVT variations for the m-CMVD circuit at three worst-case corners. Here, the *x*-axis indicates the time (in nanoseconds or ns) during the transient simulations, while the *y*-axis represents the combined currents flowing through the VCSEL diode at different temperatures from −55 °C to 125 °C. The wide range of modulation currents of m-CMVD (0.1–10 mA_pp_) shown in these results demonstrates its ability to adapt to various indoor distance conditions, particularly in specialized applications such as monitoring the fall accidents of elderly individuals. This current range effectively covers the distances typically encountered in indoor environments, ensuring reliable performance under changing conditions.

Figure 10 shows the chip layout for the proposed oRx, where the top APD detects the light signal and converts it to DC current for the concurrent APC operation. The middle APD generates the photocurrent from the reflected light, while a dummy APD is positioned below it to maintain symmetry in the circuit. The chip core of the oRx occupies an area of 211 × 176 μm^2^.

## 4. Measured Results

The proposed OTRx chips were fabricated by using a 180 nm CMOS process. Figure 11 shows the chip photo along with test setup, where the m-CMVD and the APC channel use a 3.3 V supply, while the oRx uses a 1.8 V supply. For testing, a 3.3 V_pp_ pulse input with a 50 ns pulse width (V2) is applied to the m-CMVD. Then, the oRx and the APC channels receive incoming pulses. Then, the output pulses of the oRx are measured using an oscilloscope. The core occupies an area of 377 × 348 µm^2^, consuming 134 mW in maximum.

Figure 12 demonstrates the measured pulse response of the proposed m-CMVD circuit, in which the output current of 1 mA is transmitted with a pulse width of 50 ns when all the switches are turned on.

Figure 13 shows the measured eye-diagrams of the proposed m-CMVD circuit, in which the output current of 1 mA is observed at the data rate of 10 Mb/s.

Figure 14 demonstrates the measured eye-diagram results of the proposed oRx at the OUT+ node with an input current of 290 µA_pp_ at various data rates of 100 Mb/s, 1 Gb/s, 1.5 Gb/s, and 2.5 Gb/s, respectively. It is clearly seen that the eye-amplitudes remain barely changed even with the increase in data rates, thus confirming consistent signal integrity across various transmission speeds.

Figure 15 presents the measured pulse response of the proposed oRx with the different input currents ranging from 100 µA_pp_ to 600 µA_pp_, where symmetric and stable differential outputs are observed at the output nodes, i.e., OUT+ and OUT−, which is attributed to the proposed FD-TIA configuration equipped with a dummy APD.

Finally, experiments with the APC mechanism were conducted. To facilitate the measurements, we have assumed that the distance to targets is fixed while the bias current of the Tx is decreasing, during which time the average strength of the input optical signals is reduced. Then, the APC channel in the oRx detects the reduced bias current and then recovers (or increases) the bias current higher than the reference bias current (1 mA).

Figure 16 demonstrates the measured bias current of the APC channel, when the decreasing input current is detected in the range between 3–250 µA_pp_. Then, it is clearly observed that the DC level increases as the detected strength of the input signal decreases. These results indicate that the APC channel can effectively compensate for the variation in the bias current when the Tx provides the degraded bias current, thus confirming the potential of the proposed APC mechanism.

Table 1 compares the performance of the proposed m-CMVD with previously reported CMOS VCSEL drivers. Ref. [22] described a current-mode VCSEL driver implemented in a 130 nm CMOS process, featuring a digital APC with time division sensing and supporting bias currents up to 20 mA. However, it suffered from high power consumption and a relatively large chip area. Similarly, Ref. [23] introduced a multi-rate VCSEL driver with a digital APC utilizing a flash-SAR ADC in a 130 nm CMOS process. While it offered advanced calibration capabilities and supported high modulation currents (5–20 mA_pp_), its total power consumption reached 471.36 mW and the chip area was very large. Meanwhile, Ref. [24] proposed a current-mode VCSEL driver designed in a 65 nm CMOS process, which provided modulation currents up to 14 mA_pp_. Although it used a charge pump with a feedback loop for APC to achieve effective current control, it consumed 376 mW of power, making it less suitable for low-power applications.

This work presents a modified single-ended m-CMVD with an innovative APC implementation which exploited the APD feedback. The proposed design achieves a stable bias current of 0.93–1.42 mA and supports variable modulation currents up to 10 mA_pp_ with a significantly reduced power consumption of 11 mW. Furthermore, the compact chip area of 0.131 mm^2^ highlights the advantages of integrating APC and VCSEL driver circuits into a cost-effective 180 nm CMOS process.

## 5. Conclusions

We have presented a novel optoelectronic transceiver (OTRx) realized in a 180 nm CMOS process for the applications of short-range indoor monitoring LiDAR sensors. The OTRx includes an m-CMVD circuit as a Tx and an oRx circuit as a Rx along with a concurrent APC channel using three on-chip APDs. The specially dedicated APC channel suggests a unique design methodology to compensate for the bias current of the Tx concurrently. Chip measurements confirmed that the m-CMVD produces a stable output of 1 mA, and a power consumption of 11 mW with a 3.3 V supply, while the oRx demonstrates reliable eye-openings up to 2.5 Gb/s with a power consumption of 48 mW with a 1.8 V supply. The APC channel can detect the degraded average optical power of the m-CMVD and then generate a DC bias current ranging from 0.93 mA to 1.42 mA when the input signal amplitude decreases from 250 µA_pp_ to 3 µA_pp_. In comparison to previous works, the proposed OTRx achieves significantly lower power consumption and a more compact chip area. The innovative APC implementation, located on the receiver side using an on-chip APD, eliminates the need for external components, further enhancing its cost-effectiveness for short-range sensor applications. This ultimately demonstrates its potential as a low-cost solution in the short-range LiDAR and optical sensing system applications.

## Figures and Tables

**Figure 1 sensors-25-00753-f001:**
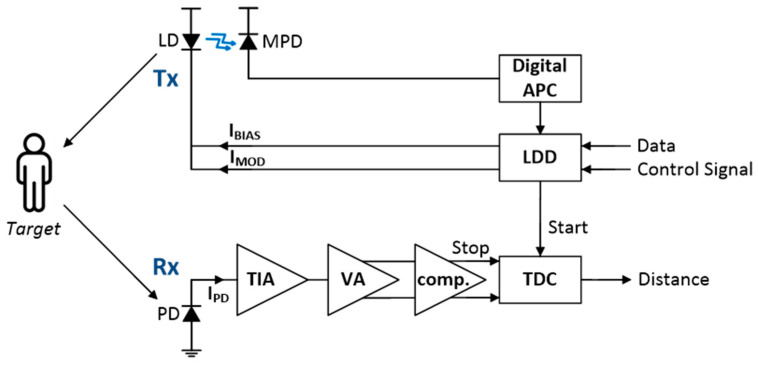
Block diagram of a typical LiDAR sensor [6,7].

**Figure 2 sensors-25-00753-f002:**
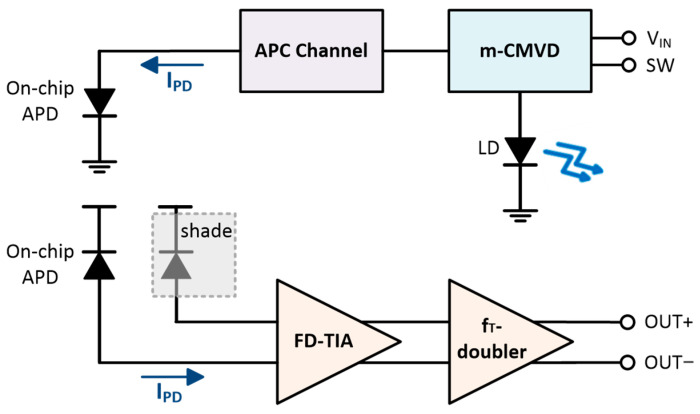
Block diagrams of the proposed OTRx.

**Figure 3 sensors-25-00753-f003:**
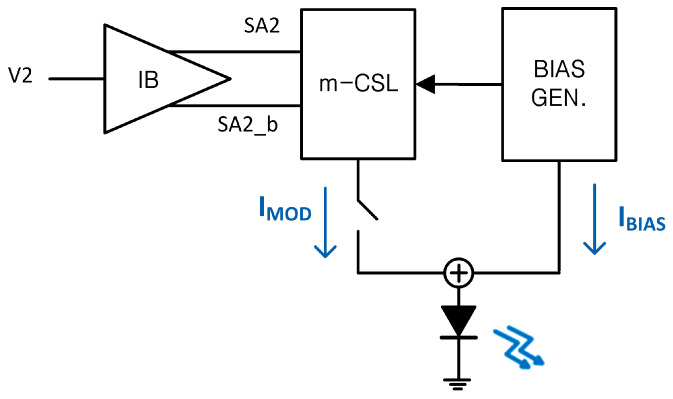
Block diagram of the proposed m-CMVD.

**Figure 4 sensors-25-00753-f004:**
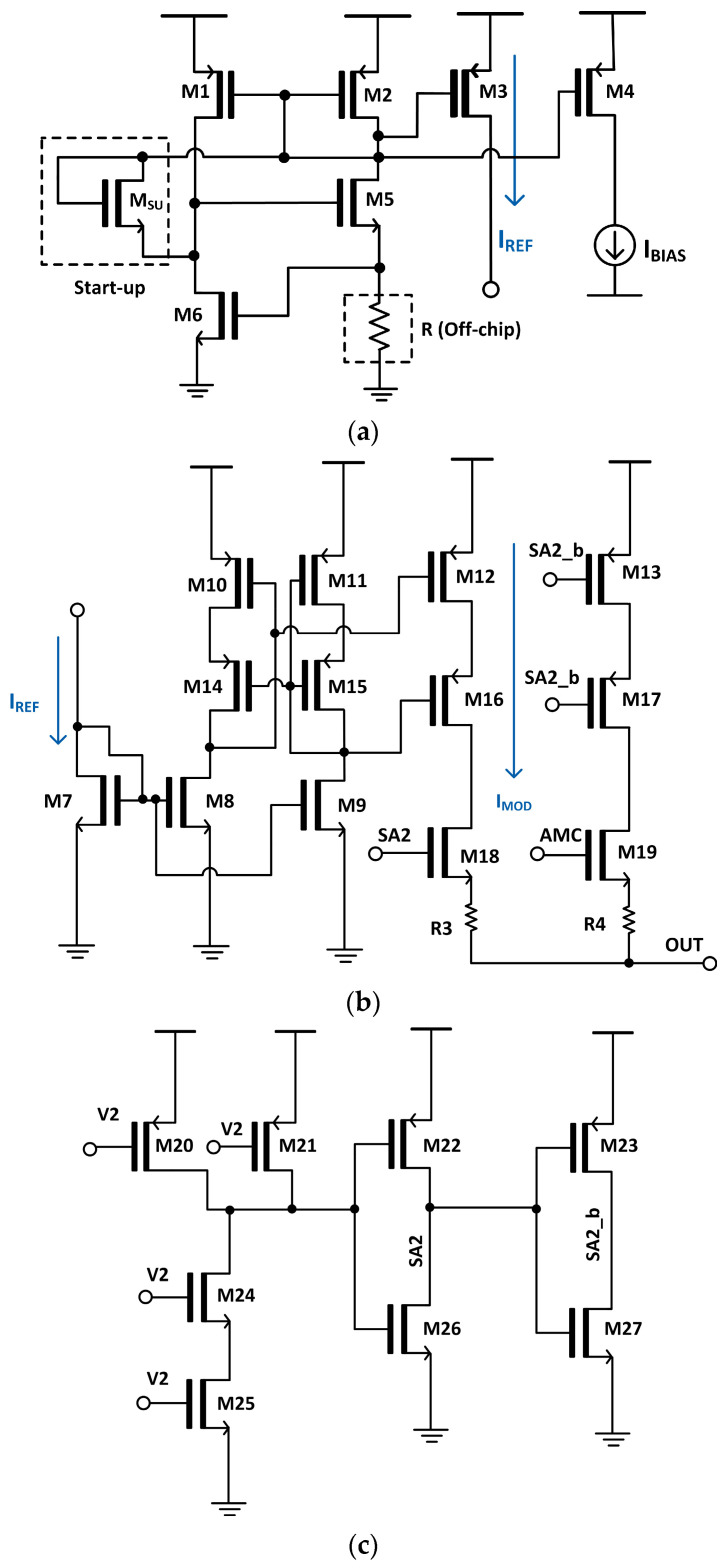
Schematic diagrams of the (**a**) bias generator, (**b**) m-CSL circuit, and (**c**) IB.

**Figure 5 sensors-25-00753-f005:**
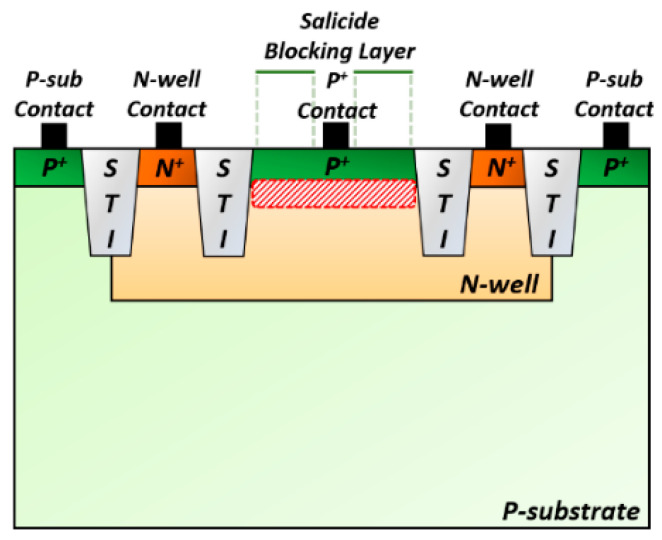
Cross-section view of the on-chip P^+^/NW APD.

**Figure 6 sensors-25-00753-f006:**
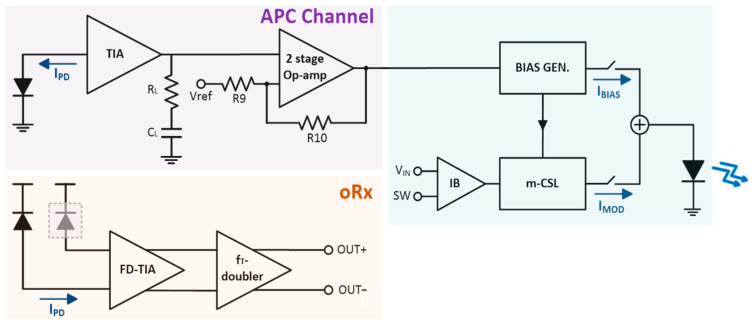
Block diagram of the proposed optoelectronic receiver (oRx).

**Figure 7 sensors-25-00753-f007:**
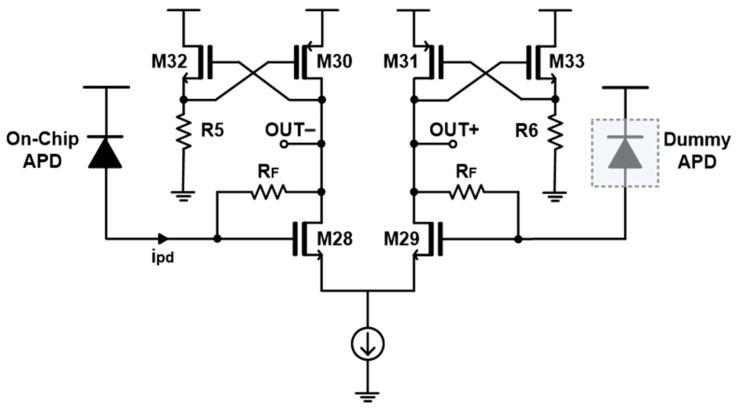
Schematic diagram of the proposed FD-TIA.

**Figure 8 sensors-25-00753-f008:**
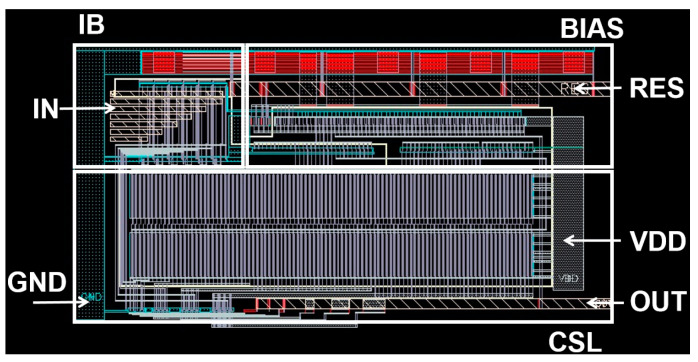
Chip core layout of the m-CMVD circuit.

**Figure 9 sensors-25-00753-f009:**
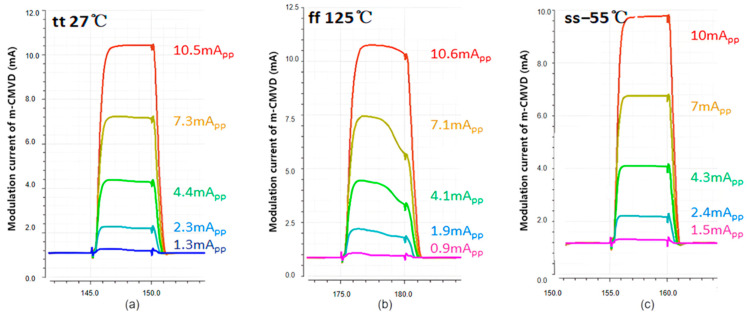
Simulated PVT variations in the m-CMVD with the combined currents of 0.1–10 mA_pp_ for a pulse-width of 5 ns: (**a**) TT, 27 °C, 3.3 V, (**b**) FF, 125 °C, 3.63 V, and (**c**) SS, −55 °C, 2.97 V.

**Figure 10 sensors-25-00753-f010:**
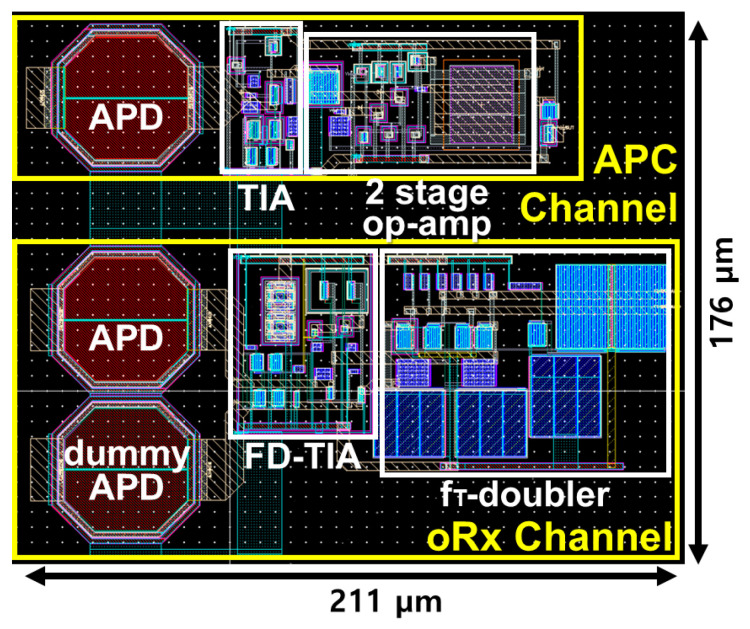
Layout of the proposed oRx.

**Figure 11 sensors-25-00753-f011:**
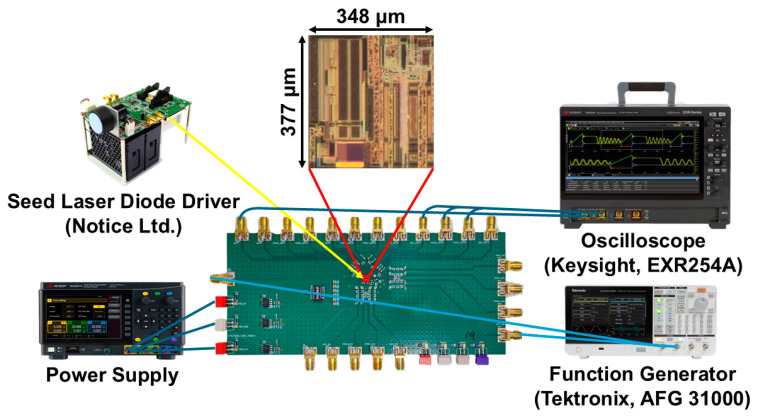
Chip photo of the proposed OTRx chip and its test setup.

**Figure 12 sensors-25-00753-f012:**
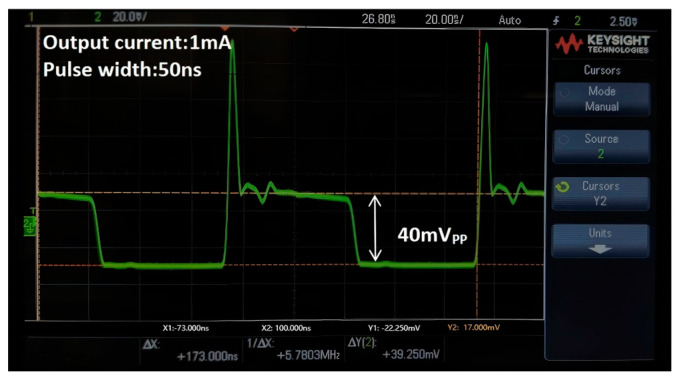
Measured transient responses of the proposed m-CMVD.

**Figure 13 sensors-25-00753-f013:**
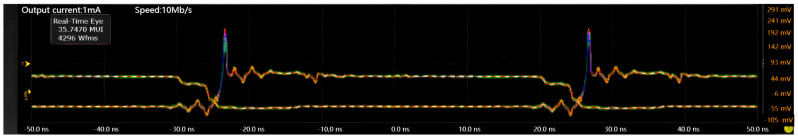
Measured eye-diagrams of the proposed m-CMVD.

**Figure 14 sensors-25-00753-f014:**
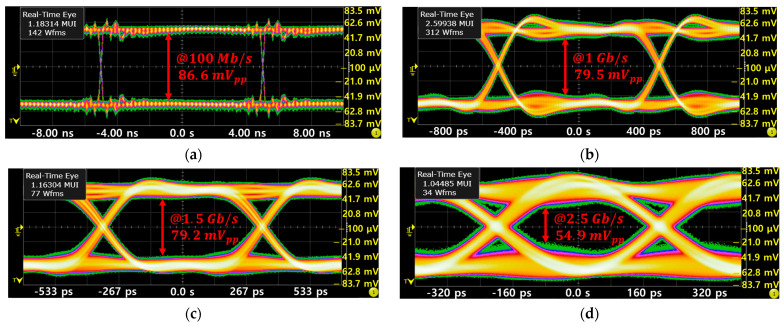
Measured eye-diagram of the proposed oRx for 290 µA_pp_ input current at different data rates of (**a**) 100 Mb/s, (**b**) 1 Gb/s, (**c**) 1.5 Gb/s, and (**d**) 2.5 Gb/s, respectively.

**Figure 15 sensors-25-00753-f015:**
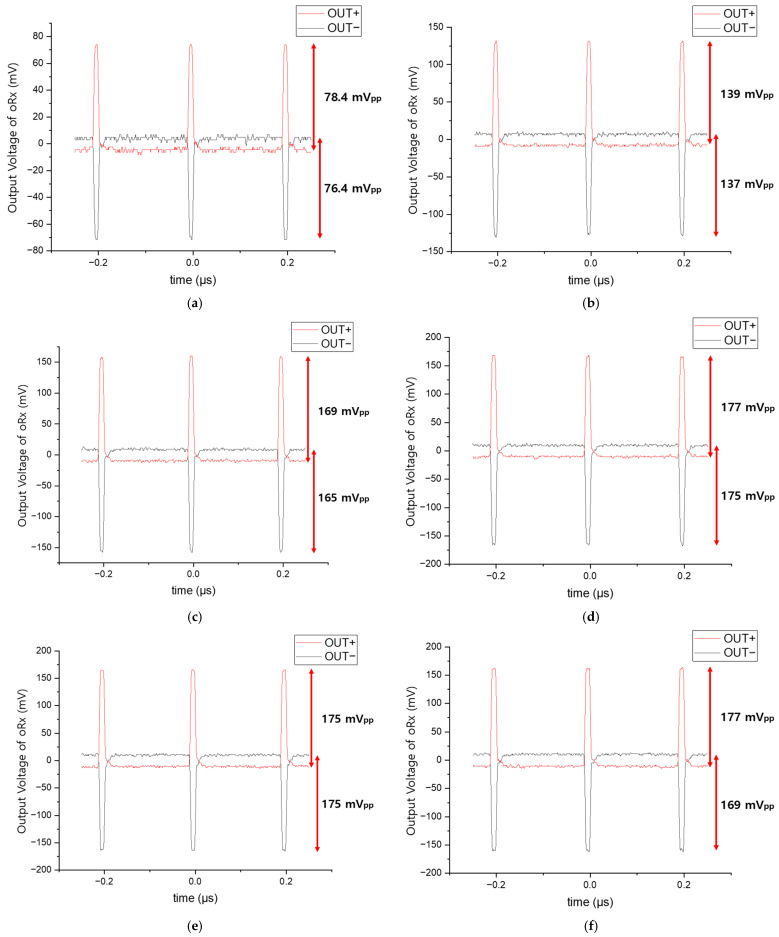
Measured pulse response of the proposed oRx for the different input currents of (**a**) 100 µA_pp_, (**b**) 200 µA_pp_, (**c**) 300 µA_pp_, (**d**) 400 µA_pp_, (**e**) 500 µA_pp_, and (**f**) 600 µA_pp_, respectively. (pulse width: 10 ns).

**Figure 16 sensors-25-00753-f016:**
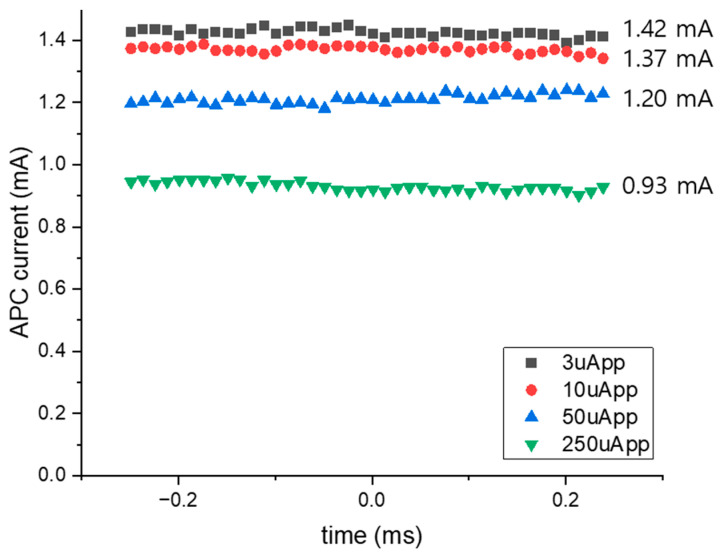
Measured bias current of the proposed OTRx with respect to the degraded input current.

**Table 1 sensors-25-00753-t001:** Performance comparison with the previously reported CMOS VCSEL drivers.

Parameters	[22]	[23]	[24]	This Work
CMOS Process (nm)	130	130	65	180
Optical Device	VCSEL	VCSEL	VCSEL	VCSEL
Architecture	current-mode	current-mode	current-mode	current-mode
Signaling Configuration	single-ended	differential	single-ended	single-ended
Driver Type	common-cathode	common-anode	common-cathode	common-cathode
Supply Voltage (V)	3.3	1.2/3.3	3.3	3.3
Bias Current (mA)	5–20	5–20	-	1
Maximum Modulation Current (mA)	5–20	5–20	14	10
Max. Power Consumption (mW)	371	471.36	376	11
APC Implementation	Digital APC with Time Division Sensing	Digital APC with Flash-SAR ADC	Charge Pump with feedback loop	Analog APC with APD feedback
APC Range(mA)	5–20	5–20	-	0.93–1.42
Chip Area(mm^2^)	4.823	4.2	3	0.131

## Data Availability

Data are contained within the article.

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
