# Peer review of "A CMOS Optoelectronic Transceiver with Concurrent Automatic Power Control for Short-Range LiDAR Sensors"

_sensors, 2025, doi:10.3390/s25030753_

Round 1

Reviewer 1 Report

Comments and Suggestions for Authors

This paper presents an optoelectronic transceiver (OTRx) implemented using 180nm CMOS technology for short-range LiDAR sensor applications. The proposed OTRx utilizes three on-chip APDs and includes an m-CMVD circuit in the Tx along with an APC channel. The authors validated this through simulation, fabrication, and measurement that the OTRx generates the necessary bias current in the APC channel to produce stable outputs from the m-CMVD. This paper shows the potential of the OTRx as a low-cost solution for short-range LiDAR sensor applications. However, it is recommended to revise the paper to address the following comments, which will make the paper more robust.

References :

• A minimum of 20 references is considered appropriate for the ‘References’ section.

- For the magazine’s style and the article's content, I consider an appropriate number of references to be more than 20. Select them carefully to benchmark your work against the state of the art effectively.

Line 45-49 :

  Background of the research

-The previous researches or the comparable researches did not introduced. What are the other researches of low-cost and low-power applications based on VCSELs that could further strengthen the argument, along with references?

Line 66-70 :

  Block diagram of the proposed m-CMVD.

- The more detailed explanation for Figure 3 is necessary. For example, How IMOD and IBIAS in the figure operate as a current mirror?

Line 106-115 :

  Cross-section view of the on-chip P+/NW APD.

- It is unclear what point you want to explain through the on-chip APD. Details are needed on how the proposed on-chip APD differs from a typical APD and how it is connected to the FD-TIA, including a detailed cross-section.

Line 183 :

• There is an issue with the resolution of Figure 9.

- The resolution of the figure is not appropriate. Please re-upload the figure with clearly visible x-axis and y-axis.

Line 193-196 :

• ‘Measured Results’ sections need to be improved in more details for the future readers.

- For Figure 11, please add an explanation of the roles and conditions of each component in the test setup, as well as how the measurements were conducted.

Line 214-217 :

• There is an issue with the resolution of Figure 12-14.

- Figures 12–14 need to be regenerated as new graphs based on the measurement results. If this is difficult, please improve the resolution so that the x-axis and y-axis are clearly visible.

Line : 175-180

modulation currents(mApp)

- A clearer explanation of modulation currents (mApp) is necessary to avoid confusion. Please label the mApp in the transmitter and the mApp in the receiver with different names. It is difficult to understand what advantages are being demonstrated.

Line : 236-250

• Conclusions

- A comparison table or analysis comparing this work with the previously published works of the same area is needed.

Comments on the Quality of English Language

The manuscript seems to require English language revision.

The following sentence uses numerous modifiers, making it complex. It is recommended to break it into shorter sentences for clarity. Also, the grammar should be checked. For examples,

Line : 126-129 :

Fig. 6 illustrates the block diagram of the proposed optoelectronic receiver (oRx), where an on-chip APD receives the reflected lights from target objects and generates an input photocurrent (ipd), a dummy APD is added at the differential input node to discard a usually inserted passive capacitor that is electrically modeling the on-chip APD.

Line : 129-133 :

This dummy APD alleviates the DC offset issue occurred between the two differential outputs and helps to produce symmetrical outputs through this balanced architecture. Therefore, this oRx can inherently avoid the need of an offset cancellation circuit, which results in reduced circuit complexity and smaller chip area.

Author Response

1. References: A minimum of 20 references is considered appropriate for the ‘References’ section. For the magazine’s style and the article's content, I consider an appropriate number of references to be more than 20. Select them carefully to benchmark your work against the state of the art effectively.

 (ans.) We have selected 24 references relevant to this work.

2. Line 45-49: Background of the research. The previous researches or the comparable researches were not introduced. What are the other researches of low-cost and low-power applications based on VCSELs that could further strengthen the argument, along with references?

(ans.) We have added more references on low-cost and low-power applications based on VCSELs in Section 2 ‘Background of the Research (line 42-50). Specifically, we have highlighted the advantages of VCSELs in cost-sensitive and low-power applications, as well as recent advancements in VCSEL-based technologies.

3. Line 66-70: Block diagram of the proposed m-CMVD. The more detailed explanation for Figure 3 is necessary. For example, How IMODand IBIASin the figure operate as a current mirror?

(ans.) We have revised the manuscript (Section 2, line 92-103) to include more detailed explanation of how IMOD and IBIAS operate as a current mirror in the circuit. Specifically, we have elaborated on how the reference current (IREF) generated by the bias generator (shown in Figure 4a) is mirrored and transferred through the NMOS transistors in the m-CSL circuit as shown in Figure 4b. In addition, we have included details on how adjusting the aspect ratio (W/L) of the transistors enables the precise scaling of the mirrored current. These provide flexibility in the design of the modulation current (IMOD) to meet the specific requirements of the VCSEL driver.

4. Line 106-115: Cross-section view of the on-chip P+/NW APD. It is unclear what point you want to explain through the on-chip APD. Details are needed on how the proposed on-chip APD differs from a typical APD and how it is connected to the FD-TIA, including a detailed cross-section.

(ans.) When off-chip APDs are utilized, signal distortions may occur because of the inevitable bond-wire inductance followed by the parasitic capacitance from the ESD protection diode. It may also cause difficulties in PCB design for testing and increase the implementation cost especially for the cases of multi-channel receiver arrays. Therefore, we have eliminated these effects in this work by exploiting on-chip APDs, simultaneously offering merits such as small size, low cost, and facile design.

When light enters the optical window of the APD, electron-hole pairs are generated at the junction between the P+ region and the N-well. The generated holes, under the influence of the reverse bias voltage applied to the N-well, create additional electron-hole pairs, thereby triggering the avalanche effect [14]. Then, the generated photocurrents flow into the input node of the FD-TIA that is connected to the central P+ contact of the optical window.

5. Line 183: There is an issue with the resolution of Figure 9. The resolution of the figure is not appropriate. Please re-upload the figure with clearly visible x-axis and y-axis.

(ans.) We have improved the resolution of Figure 9.

6. Line 193-196: ‘Measured Results’ sections need to be improved in more details for the future readers. For Figure 11, please add an explanation of the roles and conditions of each component in the test setup, as well as how the measurements were conducted.

(ans.) The m-CMVD and the APC channel use a 3.3-V supply while the oRx uses a 1.8-V supply. For testing, a 3.3-Vpp pulse input with a 50-ns pulse width (V2) is applied to the m-CMVD. Then, the oRx and the APC channels receive the incoming pulses. Then, the output pulses of the oRx are measured by using an oscilloscope.

7. Line 214-217: There is an issue with the resolution of Figure 12-14. Figures 12–14 need to be regenerated as new graphs based on the measurement results. If this is difficult, please improve the resolution so that the x-axis and y-axis are clearly visible.

(ans.) Figure 12-14 are regenerated in the revised manuscript.

8. Line 175-180: modulation currents(mApp). A clearer explanation of modulation currents (mApp) is necessary to avoid confusion. Please label the mApp in the transmitter and the mAppin the receiver with different names. It is difficult to understand what advantages are being demonstrated.

(ans.) We have revised Figure 9 to avoid confusion. We have clearly labeled the modulation currents in the transmitter.

9. Line 236-250: Conclusions. A comparison table or analysis comparing this work with the previously published works of the same area is needed.

(ans.) We have added a comparison table (i.e., Table 1) of this work with previously published research.

10. Line 126-129: Fig. 6 illustrates the block diagram of the proposed optoelectronic receiver (oRx), where an on-chip APD receives the reflected lights from target objects and generates an input photocurrent (ipd), a dummy APD is added at the differential input node to discard a usually inserted passive capacitor that is electrically modeling the on-chip APD.

(ans.) We have revised the sentence for better clarity as follows:

“Figure 6 illustrates the block diagram of the proposed optoelectronic receiver (oRx), where the on-chip APD detects the reflected light from a target and generates an input photocurrent (ipd). Also, a dummy APD is added at the differential input node to eliminate the need of a passive capacitor typically used to electrically model the on-chip APD.”

11. Line 129-133: This dummy APD alleviates the DC offset issue occurred between the two differential outputs and helps to produce symmetrical outputs through this balanced architecture. Therefore, this oRx can inherently avoid the need of an offset cancellation circuit, which results in reduced circuit complexity and smaller chip area.

(ans.) We have revised this sentence as follows:

“This dummy APD helps address the DC offset issue that occurs between the two differential outputs. By using this balanced architecture, it ensures the production of symmetrical outputs. Namely, this oRx architecture can inherently avoid the need of an offset cancellation circuit, thus resulting in reduced circuit complexity and small chip area.”

Reviewer 2 Report

Comments and Suggestions for Authors

Yejin Choi et al., “A CMOS Optoelectronic Transceiver with Concurrent Automatic Power Control for Short-Range LiDAR Sensors 

MDPI Sensors

General Observations

An optoelectronic transceiver is described, realized in a 180-nm CMOS technology for the applications of short-range LiDAR sensors. A modified current-mode single-ended VCSEL driver (m-CMVD) is exploited as a transmitter, while a voltage-mode fully differential transimpedance amplifier (FD-TIA) is employed as a receiver.

Specific comments

Line 21

»...emphasizing its potential…«

Use a more appropriate verb.

Line 27 

»Particularly, a…«

Eliminate particularly.

Line 34

»...circuits, thus being responsible for…«

Eliminate, »thus being responsible«

Line 35 

»...objects. Especially, the Tx…«

Eliminate »especially«.

Line 42

»Since LDs often suffer from high cost and significant bias voltage requirements, vertical-cavity-surface-emitting laser (VCSEL) diodes…«

Start sentence with »Vertical«.

Replace »since« with a more appropriate word, such as »because.

Line 46 

»Although a traditional LD driver with narrow high- power pulses remains still preferable for long-range LiDAR systems…«

Eliminate or provide reference. This is probably your conclusion?

Line 50

»...and therefore mandates an optical splitter (or coupler) to monitor the emitter optical power…«

»Therefore« is placed between two comas.

Replace »and«.

The subject of the sentence that follows and is not provided.

Replace »emitter« with »emitted«.

Line 52 

»...a novel circuit topology is necessary to solve this tricky issue….«

Please describe clearly »This tricky issue«. 

A topology does not solve an issue.

Line 58

»Particularly, the m-CMVD in Tx…«

Eliminate »particularly«.

Line 59 

»...and effectively compensate the well-known variations caused by temperature…«

»S« is missing in the third person singular.

Please provide reference for the »well-known variations«.

Line 66 and elsewhere

»Fig. 3 depicts…«

At a beginning of a sentence, replace Fig with Figure.

Line 67 

»Thus, the m-CMVD ensures reliable and consistent optical performance…«

Eliminate “thus«.

Line 68

»...and effectively compensate the well-known variations caused by temperature and device aging…«

»S« is missing in the third person singular.

Repetition of phrase »the well-known variations caused by temperature and device aging« is unpleasant, not to say disturbing.

Line 72 

»...the performance of the emitted optical pulses…«

Please explain the concept of »the performance of the emitted optical pulses«.

Line 73 

»...to mitigate this issue, an…«

Please describe the issue to mitigate.

Line 74 

»...Therefore, the APC circuit is required to monitor the output power of a laser diode by utilizing a feedback loop and adjust the bias currents accordingly to stabilize the output power….«

Eliminate »is required«.

A lot of unnecessary verbiage. Is artificial intelligence acknowledged somewhere?

Line 103 and elsewhere

»...which...«

“That” is preferred in English.

Line 109 

»..., and the optical window has a dimension of 40 µm.«

Cut a new sentence and eliminate »and«.

Do you mean dimension or thickness/ depth?

Line 112 and elsewhere

»...through which...«

The use of which requires a coma.

Decrease the use of »which«, which in English is irritating.

Line 236 

»Conclusions«

Replace Conclusions with Summary.

General Comments

This is a valuable contribution. It could be published with minor modifications without need for my re-review.

Recommendation

Publish with minor optional modifications.

Additional

Methodology of the study

The presented methodology section fails to delineate the specific selection method. It rather emphasizes the engineering implementation of an improved technique. There are no actual performance comparisons between the previously established methods and the currently introduced technique, although the reader is left with the knowledge that the state-of the art is indeed being implemented. In the opinion of this reviewer, the implementation of the most novel techniques, therefore, makes the paper sufficiently solid and rigorous to allow its publication.

Conclusions

There are no occlusions presented in this MS, just a summary of work. A comparison with the previous techniques in either speed or data throughput would significantly enhance the value of this MS to the readership. This would be particularly useful to less knowledgeable engineering readers or people outside engineering fields. It would make this article significantly more appealing to a wide field of readership. Considering that this is a rather engineering MS, this reviewer finds the absence of strong conclusions still acceptable, considering that the practicing engineers would understand its contributions.

Author Response

1. Line 21 »...emphasizing its potential…« Use a more appropriate verb.

 (ans.) We have replaced ‘emphasizing its potential’ with ‘highlighting its suitability’.

2. Line 27 »Particularly, a…« Eliminate particularly.

(ans.) We have removed ‘Particularly’.

3. Line 34 »...circuits, thus being responsible for…« Eliminate, »thus being responsible«

(ans.) We have removed ‘thus being responsible’.

4. Line 35 »...objects. Especially, the Tx…« Eliminate »especially«.

(ans.) We have removed ‘especially’.

5. Line 42 » Since LDs often suffer from high cost and significant bias voltage requirements, vertical-cavity-surface-emitting laser (VCSEL) diodes…« Start sentence with »Vertical«. Replace »since« with a more appropriate word, such as »because.

(ans.) We have updated the paragraphs as follows:

“Previous research has primarily focused on LD drivers for long-range LiDAR systems that require narrow, high-power optical pulses [8]. As an example, Ref. [9] introduced a high repetition rate CMOS driver for generating high-energy sub-nanosecond laser pulses in SPAD-based time-of-flight range finding. Although these advancements have significantly improved the LD driver design, laser diodes often face challenges due to high costs and substantial bias voltage requirements. Vertical-cavity-surface-emitting laser (VCSEL) diodes, in contrast, offer a more affordable and energy-efficient alternative with benefits such as low bias voltage and cost. As highlighted in Ref. [10], VCSELs have been adopted in cost-sensitive applications.”

6. Line 46 » Although a traditional LD driver with narrow high- power pulses remains still preferable for long-range LiDAR systems…« Eliminate or provide reference. This is probably your conclusion?

(ans.) We have provided more references.

7. Line 50 »...and therefore mandates an optical splitter (or coupler) to monitor the emitter optical power…« »Therefore« is placed between two comas. Replace »and«. The subject of the sentence that follows and is not provided. Replace »emitter« with »emitted«.

(ans.) We have revised that sentence as below:

“Yet, it is well known that VCSEL diodes are unidirectional and thus mandate the use of an optical splitter (or coupler) to monitor the emitted optical power for APC operations.”

8. Line 52 »...a novel circuit topology is necessary to solve this tricky issue….« Please describe clearly »This tricky issue«. A topology does not solve an issue.

(ans.) We have removed the sentence to avoid confusion.

9. Line 58 »Particularly, the m-CMVD in Tx…« Eliminate »particularly«.

(ans.) We have eliminated ‘particularly’.

10. Line 59 »...and effectively compensate the well-known variations caused by temperature…« »S« is missing in the third person singular. Please provide reference for the »well-known variations«.

(ans.) We have provided a reference [11] for the ‘well-known variations’, as below:

“The m-CMVD in Tx ensures reliable and consistent optical performance and effectively compensates for the well-known variations caused by temperature and device aging [11].”

11. Line 66 and elsewhere »Fig. 3 depicts…« At a beginning of a sentence, replace Fig with Figure.

(ans.) We have replaced ‘Fig.’ with ‘Figure’.

12. Line 67 »Thus, the m-CMVD ensures reliable and consistent optical performance…«

Eliminate “thus«.

(ans.) We have removed ‘thus’.

13. Line 68 »...and effectively compensate the well-known variations caused by temperature and device aging…« »S« is missing in the third person singular. Repetition of phrase »the well-known variations caused by temperature and device aging« is unpleasant, not to say disturbing.

(ans.) We have removed the repetition of the phrase.

14. Line 72 »...the performance of the emitted optical pulses…« Please explain the concept of »the performance of the emitted optical pulses«.

(ans.) We have explained the concept of the performance of the emitted optical pulses as follows:

“However, the threshold current of VCSEL diodes may increase due to the temperature variation. This increase of threshold current can lead to performance degradation issues, such as amplitude variation, increased timing jitter, and reduced bandwidth. These issues can ultimately deteriorate the reliability of VCSEL diodes.”

15. Line 73 »...to mitigate this issue, an…« Please describe the issue to mitigate.

(ans.) We have described the issue as follows:
“However, the threshold current of VCSEL diodes may increase due to the temperature variation. This increase of threshold current can lead to performance degradation issues, such as amplitude variation, increased timing jitter, and reduced bandwidth. These issues can ultimately deteriorate the reliability of VCSEL diodes. Automatic power control (APC) mechanism is employed to mitigate this issue in general, thereby compensating for the temperature-induced rise of threshold current. Therefore, the APC circuit monitors the output power of a VCSEL diode by utilizing a feedback loop and adjusts the bias current accordingly to stabilize the output power.”

16. Line 74 »...Therefore, the APC circuit is required to monitor the output power of a laser diode by utilizing a feedback loop and adjust the bias currents accordingly to stabilize the output power….« Eliminate »is required«. A lot of unnecessary verbiage. Is artificial intelligence acknowledged somewhere?

(ans.) We have revised the sentence by eliminating the phrase “is required” to make the statement more concise, as below:

“...Therefore, the APC circuit monitors the output power of a VCSEL diode by utilizing a feedback loop and adjusts the bias current accordingly to stabilize the output power....”

17. Line 103 and elsewhere »...which...« “That” is preferred in English.

(ans.) We have used ‘that’ instead of ‘which’.

18. Line 109 »..., and the optical window has a dimension of 40 µm. « Cut a new sentence and eliminate »and«. Do you mean dimension or thickness/ depth?

(ans.) We have revised the sentence and removed ‘and’.

The ‘dimension’ refers to the diameter (although the shape is octagonal). To avoid confusion, the sentence has been revised as follows:

“The optical window has a size of 40 µm x 40 µm.”

19. Line 112 and elsewhere »...through which...« The use of which requires a coma. Decrease the use of »which«, which in English is irritating.

(ans.) We have added a coma before “through” and decreased the use of ‘which’.

20. Line 236 »Conclusions« Replace Conclusions with Summary.

(ans.) We have added more descriptions for Table 1 and elaborated ‘Conclusion’.

21. Methodology of the study: The presented methodology section fails to delineate the specific selection method. It rather emphasizes the engineering implementation of an improved technique. There are no actual performance comparisons between the previously established methods and the currently introduced technique, although the reader is left with the knowledge that the state-of the art is indeed being implemented. In the opinion of this reviewer, the implementation of the most novel techniques, therefore, makes the paper sufficiently solid and rigorous to allow its publication.

(ans.) We have revised the methodology section to clearly outline the specific selection methods and provide performance comparisons between the proposed OTRx and the previously reported methods. These can be found in ‘Table 1’ and in the following discussions.

22. Conclusions: There are no occlusions presented in this MS, just a summary of work. A comparison with the previous techniques in either speed or data throughput would significantly enhance the value of this MS to the readership. This would be particularly useful to less knowledgeable engineering readers or people outside engineering fields. It would make this article significantly more appealing to a wide field of readership. Considering that this is a rather engineering MS, this reviewer finds the absence of strong conclusions still acceptable, considering that the practicing engineers would understand its contributions.

(ans.) We have revised the conclusion section by including comparison of this work with the previous techniques in terms of power consumption, chip area, and APC implementation. These updates provide a clearer understanding of the advantages of the proposed design.

Round 2

Reviewer 1 Report

Comments and Suggestions for Authors

Line 278-279

• Conclusions (Table1)

- Please update Table 1 to compare with the latest works. The most previous works in Table 1 are published over 10 years. 

Line 310-361

• References

- Please revise it according to the reference format of Sensors.

Comments on the Quality of English Language

- This sentence seems to require English revision.

This dummy APD helps address the DC offset issue that occurs between the two differential outputs. By using this balanced architecture, it ensures the production of symmetrical out-puts. Namely, this oRx architecture can inherently avoid the need of an offset cancellation circuit, thus resulting in reduced circuit complexity and small chip area.

Author Response

1. Line 278-279, Conclusions (Table1): Please update Table 1 to compare with the latest works. The most previous works in Table 1 are published over 10 years. 

(ans.) Thanks a lot for this comment. We have updated the references [22] and [24] with relatively recent works.

2. Line 310-361, References: Please revise it according to the reference format of Sensors.

(ans.) We have revised the references to the format of Sensors.

3. This sentence seems to require English revision.

This dummy APD helps address the DC offset issue that occurs between the two differential outputs. By using this balanced architecture, it ensures the production of symmetrical outputs. Namely, this oRx architecture can inherently avoid the need of an offset cancellation circuit, thus resulting in reduced circuit complexity and small chip area.

(ans.) We have revised the sentence as follows:

“This dummy APD helps to mitigate the issue of DC offset currents between the two differential out-puts by utilizing a balanced architecture, thus producing symmetrical outputs. In other words, the proposed oRx architecture can eliminate the need for an offset cancellation circuit, consequently reducing the circuit complexity and chip area.”
